# Accretion of Galaxies around Supermassive Black Holes and a Theoretical Model of the Tully-Fisher and M-Sigma Relations

## Nick Gorkavyi

Science Systems and Applications, Inc. (SSAI), 10210 Greenbelt Road, Suite 600, Lanham, MD 20706, USA; nick.gorkavyi@ssaihq.com

**Abstract:** The observed Tully-Fisher and Faber-Jackson laws between the baryonic mass of galaxies and the velocity of motion of stars at the edge of galaxies are explained within the framework of the model of accretion of galaxies around supermassive black holes (SMBH). The accretion model can also explain the M-sigma relation between the mass of a supermassive black hole and the velocity of stars in the bulge. The difference in the mechanisms of origin of elliptical galaxies with low angular momentum and disk galaxies with high angular momentum can be associated with 3D and 2D accretion.

**Keywords:** galaxies; supermassive black holes; accretion model; rotation of galaxies; Tully-Fisher law; Faber-Jackson relation; M-sigma relation

## 1. Introduction

An origin of the observed Tully-Fisher [1] and Faber-Jackson [2] relations is still unclear and is actively studied (see, for example, [3–6]). the derivation of the Tully-Fisher law $M \propto V^4$, which relates the barionic mass of a disk galaxy $M$ and the asymptotic rotation velocity $V$, is often based on unfounded assumptions. Some believe that the explanation of these relations is possible only outside the framework of the classical Newton-Einstein theory of gravity. For example, the MOND (Modified Newtonian dynamics) proposes to change the law of gravity for weak fields at the edge of galaxies in such a way that the Tully-Fisher relation holds true [7].

Some authors propose to take into account the off-diagonal elements of the metric tensor associated with the rotation of the galaxy [8,9]. The magnitude of this effect is still unclear. The non-relativistic rotation of the galaxy changes $g_{00}$ and the radial component of the gravitational acceleration (see, for example, [10]) by $(V/c)^2 \sim 10^{-6}$. The solution for the off-diagonal terms of the metric tensor depends on some assumptions that are not yet well understood. For example, in [8], the effect of rotation increased the galactic gravity by 30%. This is several times less than the required value. In [9], the effect of rotation turned out to be larger, but still about two times smaller than the observed excess of gravitational attraction in galaxies.

The Tully-Fisher and Faber-Jackson relations can be explained within the framework of the classical theory of gravity, if the role of supermassive black holes in the galaxy formation model is properly taken in account.

The hierarchical model of galaxy formation competes with the model of accretionary growth of galaxies around the SMBH. Although the SMBH mass is usually a small fraction ~0.1% of the galactic bulge mass, there is a high correlation between the SMBH mass and galaxy parameters, such as the bulge mass and stellar velocity dispersion in the bulge (M-sigma relation) [11–15]. Many authors consider these correlations as a sign that black holes (BH) may play an important role in the formation of galaxies of different types, for example, BHs and bulges may coevolve [16]. Observations confirm the presence of SMBH in almost every galaxy up to the earliest stages of the expansion of the Universe [15,16]. Models that

assume SMBHs grow after galaxy formation face difficult problems. They cannot explain the fast growth of central holes, and also contradict the existence of small galaxies with huge SMBHs, the mass of which is estimated to be half that of the bulge [17]. Alternative scenarios that assume the existence of SMBH before the formation of galaxies are considered in a number of works—see, for example, [18–21]. The concept that supermassive black holes act as seeds for the growth of galaxies is getting popular [20,22,23].

Consider the scenario of the formation of galaxies around supermassive black holes. When the age of the expanding Universe reached 380 thousand years (the epoch of recombination), the proton-electron plasma cooled down to 3000 kelvins and turned into atomic hydrogen. The gravitational Jeans instability arose in a medium of hydrogen and dark matter [24,25]. This instability is characterized by the following dispersion equation (see, for example, [24–26]):

$$\omega^2 = k^2 c_s{}^2 - 4\pi G\rho \tag{1}$$

where $\rho$ is the bulk density of the medium; $c_s$ is the speed of sound; wave vector $k = 2\pi/\lambda$, where $\lambda$ is the perturbation wavelength; $\omega = 2\pi/T$—the oscillation frequency, $T$—the oscillation period. The Jeans instability condition:

$$k^2 c_s{}^2 - 4\pi G\rho < 0 \tag{2}$$

This condition is satisfied for density perturbations with a wavelength greater than the critical one:

$$\lambda > c_s \sqrt{\frac{\pi}{G\rho}} \sim 100 \text{ ly} \tag{3}$$

where the density of gravitating matter $\rho \sim 3 \times 10^{-21} \text{g/cm}^3$ and $c_s \sim 10^6 \text{cm/s}$. The mass of the resulting cloud can be estimated as [25–27]:

$$m \approx \rho \left(\frac{\lambda}{2}\right)^3 \sim 10^5 \ M_\odot \tag{4}$$

The characteristic time for the development of the Jeans instability is:

$$T_j > \sqrt{\frac{\pi}{G\rho}} \sim 4 \times 10^6 \text{ years} \tag{5}$$

Equation (5) does not take into account the expansion of the Universe, so this characteristic time is underestimated. The cosmological environment during the era of recombination was very homogeneous, therefore, within a few million years after the Big Bang, the Universe turned into a population of identical clouds of gas, which we will call Jeans clouds. These clouds interacted with the population of supermassive black holes with masses ~$10^{6-9} \ M_\odot$. Some authors consider primordial black holes with masses up to ~$10^{6-7} \ M_\odot$ [28]. In cyclic models of the Universe the spectrum of black holes can extend up to ~$10^{8-9} \ M_\odot$ [18,21]. If the total number of supermassive holes is about $10^{11}$ (one for each galaxy), then the total mass of such SMBHs is still small part $10^{-8}$ of mass of dark matter [21]. Therefore, there are no observational restrictions on the existence of such black holes at an early stage of the expansion of the Universe. But these SMBHs can serve as seeds for the formation of galaxies. At the first stage, the SMBH captured the nearest gas cloud, forming an initial small accretion disk. This initial disk effectively accumulated the barionic matter of other clouds and grew into a protogalaxy. Let us show that the accretion model of galaxy formation around supermassive black holes can explain the enigmatic Tully-Fisher relation, as well as some other observed phenomena.

## 2. Disk Galaxies and the Tully-Fisher Relation

Note that the Tully-Fisher law is best suited not for maximum rotation velocities, but for asymptotic ones, at the edge of galaxies [29]. Therefore, the Tully-Fisher relation may depend on the conditions at the galactic edges. During the accretionary growth of a disk

(spiral or lenticular) galaxy with mass $M$, which moves in the medium of intergalactic gas, the maximum radius of the galaxy $R$ should be equal to the radius of the sphere of gravity, at the boundary of which the gravitational attraction of the galaxy becomes equal to the external perturbing force $f$ [30], caused, for example, by the gravitational influence of neighboring galaxies or intergalactic clouds:

$$\frac{GM}{R^2} = f \tag{6}$$

Expression (6) should not be considered as a requirement for a constant surface density for all galaxies. This is the condition of sufficient gravitational attraction at the edge of the protogalaxy, and it is applicable to any gravitating systems: from galactic bulges to thin disks around black holes. It can be assumed that the external force $f$ was approximately the same for all galaxies formed at an early stage in the evolution of the Universe. The magnitude of the gravitational force at the edge of our galaxy is $\frac{Gm}{R^2} \sim 2 \times 10^{-8} \frac{\text{cm}}{\text{s}^2}$ for $M \sim 10^{12}\ M_\odot$ and $R = 10^5$ ly. The circular motion velocity $V$ for clouds at the edge of the galaxy can be found from the expression:

$$V^2 = \frac{GM}{R} \tag{7}$$

Substituting radius from (7) into Equation (6), we get:

$$M = \frac{V^4}{Gf} \tag{8}$$

The mass included in Equations (6)–(8) is the mass of the gas, that is, the baryon mass. In our opinion, dark matter in the form of, for example, black holes joins galaxies later, after braking on a disk of baryonic matter. If the magnitude of the external force $f$ does not depend on the velocity, or this dependence can be neglected, then relation (8) will coincide with the famous Tully-Fisher relation between the baryonic mass of galaxies and the peripheral speed of rotation of spiral and lenticular galaxies [1]. Most often, the Tully-Fisher law is explained by assuming that the surface luminosity or surface mass density are constant for all disk galaxies [31,32]. This an unproven assumption leads to an expression like (6), but the physics of these equations is different. Our model is preferable because the physics of expression (6) depends on the gravitational attraction of the galaxy and on the environment. The most real source of an external perturbing force $f$ is the gravitational disturbances from the outer Jeans clouds with a mass $m \sim 10^5 M_\odot$:

$$f = \frac{Gm}{d^2} \sim 10^{-8}\ \text{cm/s}^2 \tag{9}$$

where $d$ is the minimum distance between a cloud (or star) at the edge of a galaxy and an intergalactic Jeans cloud. For estimation (9) it was assumed that $d = 25$ ly, or half of the typical distance between Jeans clouds (3). Thus, the estimates of the modern gravitational acceleration at the edge of the Galaxy and the gravitational perturbation from Jeans clouds at the early stages of galaxy formation (9) are in good agreement. Let us estimate the value of $f$ in another way. According to, for example, [32], the Tully-Fisher law from observational data can be written in the following form: $M = 50V^4$, where $M$ is the mass of the galaxy in solar masses, and the velocity $V$ is in km/s. If $Gf = 1/50$, then $f \sim 1.5 \times 10^{-8}$ cm/s$^2$, in good agreement with other our estimates. Since the concentration of Jeans clouds at the initial moment was the same throughout the Universe, then the distance $d$ will be the same, as well as the perturbing force $f$ for all growing galaxies. Note that the importance of gravitational perturbations of galactic stars from intergalactic clouds in the modern era was noted long ago [33,34]. The gravitational influence of Jeans clouds on growing galaxies was at its maximum during the era of galaxy formation, when the density of the medium was many orders of magnitude greater than in modern times. Later, the concentration of

intergalactic clouds fell due to the expansion of the universe and due to their capture into galaxies. At the same time, galaxies grew by accretion and the addition of dark matter $M_{DM}$. The Tully-Fisher relation (8) obtained for protogalaxies from baryonic matter $M_b$ must also evolve with time. An increase in the mass $M$ and a decrease in $f$ allows the accretion disk to occupy a large area:

$$R(t)^2 = \frac{G[M_b + M_{DM}(t)]}{f(t)} \tag{10}$$

Accordingly, the velocity at the edge of galaxies will also change:

$$V^2 = \frac{G[M_b + M_{DM}(t)]}{R(t)} \tag{11}$$

From (10) and (11) we get:

$$M_b + M_{DM}(t) = \frac{V^4}{Gf(t)} \tag{12}$$

If the accretionary growth of the galaxy and the addition of dark matter occurs in proportion to the initial baryon mass: $M_{DM}(t) = q(t)M_b$, then we obtain a generalized form of the Tully-Fisher law:

$$M_b = \frac{V^4}{Gf(t)[1 + q(t)]} \tag{13}$$

In clusters of galaxies, the concentration of Jeans clouds was higher than in the space between clusters. Expressions (8) and (9) establish a relationship between the speed of rotation of galaxies and the concentration of Jeans clouds: $V \propto f^{1/4} \propto d^{-1/2}$. This probably explains why lenticular galaxies located in the middle of clusters, where the distance between clouds $d$ was smaller, rotate faster than spiral galaxies [35–37]. Both chaotic and average relative velocities can exist between the intergalactic gas clouds and the galaxies growing around the SMBH. In the case of an average relative velocity between the SMBH and the Jeans clouds, directed, for example, along the $X$ axis, an increased amount of clouds will arrive from this direction. Obviously, clouds approaching the SMBH along the $X$ axis will be captured in the $XY$ plane, or $XZ$, or any intermediate planes, but they will not be captured in the $YZ$ plane, that is, perpendicular to the line of initial movement. It is impossible to throw a stone so that the parabola of its movement is located in a horizontal plane: it will always be vertical, and the center of the Earth will lie in the plane of the orbit. Therefore, the average speed of movement between supermassive black holes and clouds will lead to the fact that the axes of protogalaxies will be distributed anisotropically: there will be the $X$ direction, which the axes of disk galaxies will avoid. This corresponds to quadrupole anisotropy. Such an anisotropy in the distribution of the rotation axes of galaxies was discovered in the study of the catalog of radio sources [38,39] of 10 thousand galaxies with jets. The direction of these jets is close to the rotation axes of the galaxies (or the orientation of the accretion disk around the central black hole). It is shown that the axes of rotation of galaxies avoid a certain direction, that is, two opposite parts of the sky. Amirkhanyan's articles [38,39] also review previous works on this topic. The phenomenon of anisotropy of the axes of galaxies is remarkable in that it is observed for fairly close galaxies.

## 3. Elliptical Galaxies and the Faber-Jackson Relation

Faber and Jackson discovered a law for elliptical galaxies that is similar to (8) [2]. It is clear that the discussed mechanism of connection between the mass of the galaxy and the speed of peripheral motion should work not only for rotational, but also for other types of motion. The stars in an elliptical galaxy move in elliptical orbits with a high eccentricity. However, at the edge of the galaxy, they will experience a similar gravitational perturbation

$f$ from intergalactic clouds, that is, they will obey condition (6). The only difference is that to calculate the speed of bodies in elliptical galaxies, it is necessary to use not the condition of circular motion (7), but the virial theorem (for example, [40]):

$$\sigma^2 = \frac{1}{5}\frac{GM}{R} \tag{14}$$

where $\sigma$ is the one-dimensional velocity dispersion, which leads to the final expression:

$$M = 25\frac{\sigma^4}{Gf} \tag{15}$$

Thus, gravitational perturbations from Jeans clouds at the edge of galaxies during the era of their formation around supermassive black holes are the mechanism that explains both the Tully-Fisher and Faber-Jackson relations. The law (15) can be generalized in the same way as the Tully-Fisher law (13).

The reason for the formation of different types of galaxies is still not clear: elliptical galaxies, which have a small angular momentum, and disk (spiral and lenticular) galaxies, in which this angular momentum is noticeably higher (see, for example, [41]). A possible solution can be found in the accretion model of galaxy formation around the SMBH. Let us consider the process of collisional interaction between a Jeans cloud with radius $r$ and a disk with thickness $h$ and density $\rho_0$ around a supermassive black hole. The result of this collisional interaction depends on the angle at which the cloud crosses the disk. Let the gas component of the cloud have mass $m_g$ and density $\rho_g$. Let us assume that the cloud flies through the disk at an arbitrary angle $\mu$ (the zero value of this angle corresponds to a trajectory that is perpendicular to the disk plane) and effectively interacts with the disk. The mass of the gas disk $m_0$ involved in the collisional interaction with such a cloud can be estimated as follows:

$$m_0 = \rho_0\frac{\pi h r^2}{cos\mu} \tag{16}$$

The condition for the gravitational capture of a cloud after the collisional interaction can be written as:

$$\beta m_0 \gtrsim m_g \tag{17}$$

where the parameter $\beta$ depends on the conditions and geometry of the collision, between the cloud and the disk. For collisions with effective momentum exchange, $\beta$ can be noticeably less than 1, as shown by numerical calculations for the capture of passing solid bodies [42,43]. Condition (17) can be rewritten as:

$$\frac{3}{4}\beta\left(\frac{\rho_0}{\rho_g}\right)\left(\frac{h}{r}\right)\frac{1}{cos\mu} \gtrsim 1 \tag{18}$$

Let us assume that the density of the initial accretion disk that formed around the SMBH varies widely. If the disk is dense and condition (18) is satisfied for $cos\mu = 1$, then the disk captures clouds arriving from all directions. With such a three-dimensional accretion, the angular momenta of the absorbed clouds are multidirectional and mutually annihilate during averaging and relaxation. It is logical to assume that as a result of 3D accretion, elliptical galaxies are formed, which have a high density, and often a large mass, but a small specific angular momentum. In clusters of galaxies, the density of the medium is higher, so the percentage of elliptical galaxies formed during 3D accretion is also higher there. If the disk is not dense, then condition (18) is satisfied only in the case of $cos\mu \ll 1$, that is, the cloud is captured by the disk only for trajectories close to the disk plane $\mu \sim 90°$. And here there are two variants of the collision: when the direction of the passage of the cloud and the rotation of the disk coincide, and when they are opposite. The articles [42,43] numerically show that the disk effectively captures even massive objects that go around the central body in the same direction as the disk rotates. Thus, Jeans clouds can bring

significant angular momentum to the peripheral regions of the disk, causing the initial disk to expand and turn into a disk protogalaxy. The condition for the formation of spiral and lenticular galaxies can be written as:

$$\frac{3}{4}\beta\left(\frac{\rho_0}{\rho_g}\right)\left(\frac{h}{r}\right) < cos\mu < 1 \tag{19}$$

A cloud moving in the opposite direction slows down when it interacts with the disk. The angular momentum of the cloud and part of the disk annihilate, and the matter falls into the central part of the galaxy, where a bulge (a region similar in structure and rotation to elliptical galaxies) is formed. Density $\rho$ of the Jeans cloud consists of two components: gas $\rho_g \sim 15\%$ and dark matter density $\sim 85\%$, which can be stellar-mass black holes [20,44,45]. At collision of the Jeans cloud with the galactic protodisk, the gas of the cloud participates in the collision and merges with the gas of the disk, but the dark component will continue to move with some deceleration due to dynamic friction. As a result, the dark part of the globular cluster can enter an elliptical orbit and form of the halo of the galaxy. Dark globular clusters have already been discovered in the nearby giant elliptical galaxy NGC 5128 [46].

## 4. Epoch of Exponential Growth of Galaxies and the M-Sigma Relation

The M-sigma relation, which establishes the relationship between the parameters of a galaxy and its central black hole, still remains unexplained [11–15]. It follows from condition (6) that the area of the protogalactic disk is proportional to its mass: $\pi R^2 \propto M$. The accretionary growth of the mass of the galaxy $M$ directly depends on the area of the accretion disk, and, consequently, on the mass of the galaxy:

$$\frac{dM}{dt} = \rho(t)\pi R^2 V_{rel} \tag{20}$$

where the density of the Universe $\rho(t)$ depends on its size $a$.

$$\rho(t) = \rho_0\left(\frac{a_0}{a}\right)^3 = \rho_0\left(\frac{t_0}{t}\right)^{\delta} \tag{21}$$

Here $\rho_0$ and $a_0$ are the density and the radius of the Universe at the recombination epoch $t_0$. From (6), (20) and (21):

$$\frac{dM}{dt} = \gamma\left(\frac{t_0}{t}\right)^{\delta} M \tag{22}$$

where

$$\gamma = \frac{\pi G \rho_0 V_{rel}}{f} \tag{23}$$

Equation (22) leads to the exponential law of galaxy growth:

$$M = M_0 e^{\int \gamma(\frac{t_0}{t})^{\delta} dt} \approx M_0 e^{\frac{\gamma}{\delta-1}t_0} \sim 10^{2-3} M_0 \tag{24}$$

For last estimate in (24), we assumed $t \gg t_0$; an initial density of the accreting medium: $\rho_0 \sim 3 \times 10^{-21}$ g/cm$^3$; the relative velocity $V_{rel} = 300 - 400$ km/s; $f \approx 2 \times 10^{-8}$ cm/s$^2$ and the time $t_0$ is 0.38 million of years. It can be expected that relation (6) was also satisfied at the very beginning of the growth of the galactic disk, when the bulk of the mass of galaxy's embryo was contained in the SMBH. Thus, the mass $M_0$ can be interpreted as the mass of the central SMBH or the mass of the initial disk proportional to the mass of the SMBH. Based on the Equation (20), it can be assumed that after the epoch of recombination there was an epoch of exponential growth of galaxies, when over several million years the mass of the galaxy exceeded the mass of the central hole by 2–3 orders of magnitude [47].

Consequently, Equation (22) and its solution (24) relate the mass of the central hole to the mass of the surrounding bulge. Taking into account (15), we obtain the M-sigma relation: the relationship between the SMBH mass and the one-dimensional dispersion of chaotic velocities in the bulge:

$$M_0 = S\sigma^4 \tag{25}$$

where

$$S = \frac{25}{Gf} e^{-\frac{\gamma}{\delta-1}t_0} \sim 2 \times 10^{16}\, e^{-\frac{\gamma}{\delta-1}t_0} \left[ \mathrm{g}\left(\mathrm{s}^3/\mathrm{cm}^3\right) \right] \tag{26}$$

It follows from the observations that $S \approx 4 \times 10^{12}\, \mathrm{g}\left(\mathrm{s}^3/\mathrm{cm}^3\right)$. Ref. [11] Our estimates (26) coincide with the observed M-sigma law if $e^{-\frac{\gamma}{\delta-1}t_0} \sim 2 \times 10^{-4}$. It follows that the mass of the bulge of the galaxy reaches a value in $e^{\frac{\gamma}{\delta-1}t_0} \sim 5 \times 10^3 M_0$ in few millions years.

The mass included in Equations (20)–(26) is the baryonic mass. The admixture of dark matter does not play a significant role in the dynamics of the bulge, because, for example, in the bulge of the Milky Way, dark matter is only 17% [48].

We believe that the M-sigma relation is a direct consequence and proof of the accretion model of the origin of galaxies.

## 5. Conclusions

Gravitational perturbations from Jeans clouds at the edges of both disk and elliptical galaxies causes the observed Tully-Fisher and Faber-Jackson relations. The M-sigma relation was established as a result of an epoch of exponential growth of galaxies after the epoch of recombination. The dense initial disk around SMBH, turns into an elliptical galaxy as a result of 3D accretion. The non-dense disk around the SMBH is growing due to 2D accretion into a high angular momentum disk galaxy with a central bulge similar to elliptical galaxies. If there is a systematic velocity between the SMBH population and the surrounding gas, then the accretion model can shed light on the observed asymmetry in the distribution of the rotation axes of galaxies or accretion disks arounds SMBHs.

These hypotheses are supported by simple calculations, and we hope that they will initiate the development of more detailed models.

**Funding:** This research received no external funding.

**Institutional Review Board Statement:** Not applicable.

**Informed Consent Statement:** Not applicable.

**Data Availability Statement:** Not applicable.

**Acknowledgments:** The author thanks Alexander Vasilkov, Dmitry Makarov and Alexei Moiseev for helpful discussions.

**Conflicts of Interest:** The author declares no conflict of interest.

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
