# Peer review of "Accretion of Galaxies around Supermassive Black Holes and a Theoretical Model of the Tully-Fisher and M-Sigma Relations"

_galaxies, doi:10.3390/galaxies10030073_

Round 1
Reviewer 1 Report
Please see the attached

Author Response
Please, see the attachment.

Reviewer 2 Report
In this paper the author shows that gravitational perturbations from Jeans clouds at the edge of galaxies during the era of their formation around supermassive black holes are the mechanism that can explain the Tully-Fisher and Faber-Jackson relations.
The result is based on the assumption that the edges of disk and elliptical galaxies experience gravitational perturbations from Jeans clouds, which sets the relationship between the radius and mass of the growing galaxy.
The paper is well written and self consistent with a clear introduction and adequate set up. The explanations are clear and let the reader to reach the conclusion in a smooth way.
I thus consider the paper is suitable for publication in Galaxies.
Author Response
The author thanks the referee for the informative and useful review. The paper has been significantly revised.
Reviewer 3 Report
The author presents a simple theoretical approach to discuss the origin of the Tully-Fischer and M-sigma relations. The manuscript is overall well written and clear. Below, I present a few suggestions:
Page 1: Please cite also the pioneering works on the M-Sigma relation.
This is followed by two strong sentences
"This means that SMBHs play an important role in the formation
of galaxies. Observations confirm the presence of SMBH in almost every galaxy up to the earliest stages of the development of the Universe."
In the first one, a correlation between a property of the BH and a property of the galaxy does not necessarily imply that the former play an important role in the formation of galaxies. Expand on the formation of BH at early stages of the Universe (BH seeds).
Although the second sentence is correct, I would include that this refers mostly to massive galaxies with a spheroidal component.
I suggest including references to support both sentences.
Page 2:
At the end of Sec. 1. Please expand on the range of masses of BH 1E6-1E9 within a few Myr after the Big Bang. Is there any constraint from cosmological simulations and/or from observational studies?
Author Response
Please, see the attachment.

Reviewer 4 Report
In the draft for review, the author makes a first attempt to explain the Tully-Fisher and Faber-Jackson relationships within the framework of classical gravity taking into account the role of supermassive black holes and their environment. He concludes that the M-sigma relation is a direct consequence of the accretion model of galaxy formation. Although more detailed models are required, the hypothesis studied here is supported by simple calculations. I find the submitted manuscript novel and interesting at the same time, and I recommend its publication in the journal "Galaxies" in its current form.
Author Response
The author thanks the referee for the review. The paper has been revised.
Reviewer 5 Report
The essay deals with a possible explanation of the rotation curves at the edge of galaxies with supermassive black holes at their centres. The proposed explanation is based on newtonian physics. I have no objection to the publication of this work, which offers one more perspective to this important problem. However, the author should address a couple of issues before:
- One important missing work is that of Crosta et al., 2020, MNRAS 496, 2107. By using Gaia data, they have shown that the rotation curve of the Milky Way can be explained by adding off-diagonal term in the general relativity metric. There is need neither of dark matter, nor of MOND. I think that the author should add this bibliographic reference and comment it.
- The second point refers to the population of existing black holes at the age of recombination. The author assumed to be 106-9M☉. Can the author give some reference of these values? Because this distribution seems to be a little overestimated. For example, check Cappelluti et al., 2022, ApJ 926, id 205: the distribution of primordial black holes shown in Fig. 1 dropped at ~106 M☉.
Typo: page 2, row 59: units of the speed of sound should be cm/s, not cm/s3.
Round 2
Reviewer 1 Report
Thank you for taking into accout my suggestions.